# Influence of Structured Medium- and Long-Chain Triglycerides on Muscular Recovery Following Damaging Resistance Exercise

**DOI:** 10.3390/nu17101604

**Published:** 2025-05-08

**Authors:** Carina M. Velasquez, Christian Rodriguez, Kealey J. Wohlgemuth, Grant M. Tinsley, Jacob A. Mota

**Affiliations:** Department of Kinesiology and Sport Management, Texas Tech University, Lubbock, TX 79409, USAgrant.tinsley@ttu.edu (G.M.T.)

**Keywords:** recovery, triglycerides, supplements, muscle damage, medium-chain triglycerides, delayed onset muscle soreness

## Abstract

**Background/Objectives**: Structured medium- and long-chain triglycerides (sMLCT) may be a superior vehicle for medium-chain fatty acid delivery to peripheral tissues, such as skeletal muscle. Limited information is available concerning the effect of sMLCT on muscular performance or recovery after a muscle-damaging exercise protocol. The purpose of this study was to establish the effect of a novel formulation of sMLCT on muscular performance and recovery. **Methods**: Forty female adults (mean ± SD age = 22 ± 3 years; body mass index = 23.5 ± 3.4 kg/m^2^) were randomized into one of two study groups, placebo control [CON; *n* = 20] or sMLCT [*n* = 20], and completed five total visits to the laboratory. The baseline (i.e., pre-exercise) assessments of muscle performance, size, and soreness were compared to assessments immediately following exercise and 24, 48, and 72 h post-exercise. **Results**: No statistically significant condition × time interactions were noted for strength outcomes, although trends for condition × time interactions were present for torque over 25 ms (*p* = 0.06) and peak torque (*p* = 0.05). Similarly, no condition x time interactions were present for ultrasound echo intensity, the subjective ratings of soreness and pain, thigh circumference, leg volume, and vertical jump performance. **Conclusions**: Within the context of the current study, the ingestion of sMLCT did not significantly influence the rate of muscle strength recovery following muscle damaging resistance exercise.

## 1. Introduction

The ability to optimize recovery time following intense exercise sessions is an impactful area of research. For instance, when individuals undergo unaccustomed physical activity, discomfort and soreness within skeletal muscle often occurs [1]. This exercise-induced phenomenon is referred to as delayed-onset muscle soreness (DOMS) and is thought to be primarily driven by eccentric muscle damage (EMD). DOMS usually occurs one to two days following an exercise bout and is associated with an uncomfortable sensation primarily in skeletal muscle [2]. The onset of DOMS may be due to a variety of factors, including high tensile forces during eccentric muscle activity, which may damage the sarcolemma and intercellular components (i.e., connective tissue and skeletal muscle) [1]. In addition, DOMS may impair neuromuscular function following EMD, which may impact performance in strength and power exercises [1].

Nutritional strategies to improve the efficiency of muscular recovery following intense exercise have garnered much attention. Efficient muscle recovery is critical for not only athletes, but for the general population. Specifically, individuals interested in preventing injury or who are undergoing rehabilitation may be particularly interested in the role of muscle recovery on long-term performance outcomes [3]. Previous studies [4,5,6] have been conducted evaluating the use of supplementation to reduce and manage DOMS following eccentric exercise. A novel dietary approach to support tissue repair following exercise may include the supplementation of structured triglycerides [7,8]. Structured medium- and long-chain triglycerides (sMLCT) are produced through recombination of the fatty acids of medium- and long-chain triglycerides such that the resulting triglyceride molecule contains both medium- and long-chain fatty acids. The varying characteristics of fatty acids influence aspects of their physiological effects, such as the regulation of immune response in tissue [9,10]. For instance, in clinical settings, sMLCT have been reported to have favorable impacts on tissue repair [11,12,13,14,15], digestive disorders [11,12,13], and critical care patient outcomes [14,15]. Although there is limited research on sMLCT in non-clinical populations, the physiological mechanisms by which medium- and long-chain fatty acids may aid muscle tissue recovery are by promoting a positive nitrogen balance [14,15] and potential anti-inflammatory effects [15], thereby providing a potential rational for the use of sMLCT in athletic settings. While previous works have investigated related compounds’ influence on exercise performance [8], there is a lack of research on the effects of sMLCT on recovery from exercise.

Within the literature, it is well documented that eccentric muscle actions are associated with muscle damage [16,17,18]. When assessing muscle damage, the measures of muscle function (e.g., peak torque and rapid torque), performance (e.g., vertical jump), perceived soreness, and muscle morphology changes (e.g., circumferences and muscle quality) are used to provide the most effective means in evaluating the magnitude of damage induced during an EMD protocol [19,20,21,22,23]. Although prior research has examined muscle recovery following EMD, no direct evidence has been established regarding the effect of sMLCT on muscular performance or recovery after an EMD exercise protocol. Therefore, the purpose of this study was to establish the effect of a novel formulation of sMLCT on muscular performance and recovery following a high-volume EMD protocol. It was hypothesized that the participants ingesting sMLCT would have smaller decrements in muscular performance and may recover faster following EMD, as compared to participants ingesting a placebo.

## 2. Materials and Methods

### 2.1. Participants

Forty females (mean ± SD age = 22 ± 3 years; body mass index = 23.5 ± 3.4 kg/m^2^) volunteered to take part in this study. To be included in the study, participants had to be generally healthy, have a body mass index ≤ 30 kg/m^2^, a body mass between 50 and 100 kg, an age between 18 and 35, be recreationally active (defined as no more than 30 min of structured physical activity three times per week), and have performed resistance training less than three times per month over the previous six months. Participants also needed to be regularly menstruating or taking a combined oral contraceptive over the previous six months. Individuals were excluded from the study if they had a neuromuscular or metabolic disorder or a current or recent musculoskeletal injury within the past three months, were a smoker, reported recent weight changes (body mass changes > 4.5 kg in the past three months), or had changes in prescription medication use in the past month. Participants consented to abstain from the consumption of other dietary supplements, alcohol, and anti-inflammatory medications for the duration of the study. This study was conducted according to the guidelines established by the Declaration of Helsinki, and all procedures involving human subjects were approved by the Texas Tech University Institutional Review Board (IRB2023-1045).

### 2.2. Experimental Design

This study was a double-blind, randomized, placebo-controlled trial that included five visits to the laboratory (i.e., familiarization, pre/post0, post24, post48, and post72, with numbers referring to the duration since exercise, in hours). During the familiarization visit, participants completed a health, exercise status, and menstrual cycle questionnaire to verify inclusion and exclusion criteria. Stature and body mass were recorded using a stadiometer (HM200P, Charder Medical, Taichung City, Taiwan) and a calibrated scale (Model BWB-627-A, modified Tanita Corp., Issaquah, WA, USA), respectively, and body composition was assessed by dual-energy X-ray absorptiometry (Lunar iDXA, GE Healthcare, Milwaukee, WI, USA). Participants were then familiarized with the EMD exercise protocol to minimize the effect of learning with the strength testing and exercise procedures. Next, participants were randomized into one of two study groups, a placebo control [CON] or structured medium- and long-chain triglycerides [sMLCT] group. Randomization procedures were performed *a priori* using R software (v. 4.4.0; randomizeR package) following enrollment. Randomization was stratified by age (18–24 vs. 25–35), body mass (50–80 kg vs. 80–110 kg), and hormonal birth control use (oral contraceptive users vs. naturally menstruating). Each participant was then provided with the appropriate treatment. The dietary supplement contained vegetable oil (e.g., rapeseed and palm oil), vitamin E (to preserve freshness), and natural flavors. Both the active and placebo treatments were flavor-matched with a mint flavor.

Prior to the muscle-damaging exercise protocol, each participant was instructed to consume their assigned treatment (2 doses of ~10 g/day) for two days. Additionally, participants continued to consume their assigned treatment on each of the remaining four testing visit days (i.e., pre/post0, post24, post48, and post72). Therefore, participants consumed their assigned treatment for a minimum of 6 days.

To control for the potential effects of the menstrual cycle on study outcomes, all participants were scheduled to complete the muscle-damaging exercise protocol during the approximate early follicular phase of menstruation. Naturally menstruating participants were scheduled to complete the exercise protocol within seven days of the onset of menses (bleeding), counting from the first day of menstrual bleeding. Menstruation was confirmed via oral interview upon visiting the laboratory during the familiarization visit. Participants consuming oral combined contraceptives were scheduled to complete the eccentric muscle damaging protocol during the “placebo pill” week due to the lower concentration of female sex hormones during this week. Participants were instructed to bring their oral contraceptive pill pack during the familiarization visit to verify the dates of their placebo week.

### 2.3. Pre- and Post-Visit Procedures

For each of the testing visits (i.e., pre to post72), participants arrived at the laboratory at the same time of day (±~2 h) and consumed one of two daily doses of the assigned treatment. The laboratory was kept at a constant temperature (22 °C). After the muscle-damaging exercise visit, which included the pre and post0 assessments, follow-up laboratory testing was conducted at 24, 48, and 72 h post exercise (i.e., post24 to post72). Each day, the second daily dose of the assigned treatment was consumed at home by the participants.

Participants were asked to abstain from exercise or vigorous physical activity for at least two calendar days prior to the second visit, which included the muscle-damaging exercise session, as well as throughout the 3-day post-exercise monitoring period (i.e., until they completed the fifth and final study visit). Additionally, participants were instructed to maintain their normal lifestyle but to refrain from consuming any dietary supplements, medications not previously reported to the study investigators, alcohol, and anti-inflammatory or pain relief medication. Participants’ compliance with consuming the dietary supplements and adhering to these instructions was monitored by an online Survey Monkey questionnaire administered during the 6-day supplementation period. Furthermore, all participants consumed another dose (i.e., dose two) of the supplement prior to the testing visit, which was monitored by the investigators.

Prior to the muscle-damaging exercise visit, participants were instructed to complete a ≥3 h fast. At the study visit, the assigned dietary supplement and a prepackaged shake (Chocolate Protein Shake, 11.5 oz Premier Protein, Emeryville, CA, USA) were provided to participants to ensure similar nutritional intake preceding the muscle-damaging bout of exercise. During the visit, participants completed a habitual physical activity level questionnaire assessed by the International Physical Activity Questionnaire [24], a 24 h dietary recall assessed by the Automated Self-Administered Dietary Assessment Tool (ASA24; National Institutes of Health), three-dimensional optical imaging (3DO), brightness mode (B-mode) ultrasound imaging, muscle soreness ratings via visual analog scales (VASs), vertical jump testing, and isokinetic dynamometry. The 3DO, ultrasonography, VAS, vertical jump testing, and isokinetic dynamometry were completed prior to and following the muscle-damaging exercise protocol (i.e., pre/post0).

At each of the follow-up testing sessions, the assessments performed at the muscle-damaging exercise visit (i.e., pre/post0) were repeated, including the ASA24, 3DO, ultrasonography, VAS, vertical jump testing, and isokinetic dynamometry. These assessments were employed to establish changes from baseline (pre-exercise) values.

### 2.4. Three-Dimensional Optical Imaging

Prior to each of the testing sessions (i.e., pre, post24, post48, and post72) and following the eccentric muscle damaging protocol (post0), participants completed 3DO assessments to evaluate the potential changes in leg volume and thigh circumference using a commercial measuring booth that produces a three-dimensional avatar using 20 infrared depth sensors positioned at varying heights in the four corners of the booth, employing structured light technology (SS20, Sizestream, Cary, NC, USA). Prior to scanning, the device’s sensors were calibrated using a hanging panel with a checkerboard pattern. During the scan, participants were instructed to stand still, face forward, and hold two handles that promote consistent arm positioning.

### 2.5. Ultrasound Assessment

Participants underwent non-invasive imaging of the vastus lateralis (VL) on the dominant leg. Images were acquired with a B-mode ultrasound imaging device (GE Logiq E R8, GE Healthcare, Milwaukee, WI, USA) in conjunction with a multifrequency linear array probe (L4-12T, 5–13 MHz, 38.4 mm field of view, GE Healthcare, Milwaukee, WI, USA). The VL was marked at the proximal and distal musculo-tendon junctions, which were confirmed via B-mode ultrasound. Muscle length was quantified with a flexible tape measure to the nearest 0.5 cm between the musculo-tendon junctions. Cross-sectional area (CSA) scans were taken in the sagittal plane at 25%, 50%, and 75% of the muscle length for the VL. To capture muscle architecture, a longitudinal image was taken by scanning the entire length of the VL (i.e., proximal to distal musculo-tendon junctions) while using the extended field of view function. A generous amount of water-soluble acoustic coupling gel (Aquasonic 100 ultrasound transmission gel, Parker Laboratories, Fairfield, NJ, USA) was applied to the skin to enhance image acquisition. At each visit, two scans were taken at each scanning site, and the best image was used for analysis for that testing day.

### 2.6. Ultrasound Analysis

All ultrasound images were analyzed in an open-source imaging software (ImageJ, version 1.54; National Institute of Health, Bethesda, MD, USA). All images were scaled from pixels to centimeters prior to analysis. The polygon function was used to determine the CSA; specifically, an experienced investigator selected the region of interest, including as much of the VL as possible and excluding the surrounding fascia. Using the same selected area of interest, echo intensity was assessed by a computer-aided gray-scale analysis using the histogram function (range 0–255 a.u.). The mean echo intensity was corrected for subcutaneous fat thickness [25]. For muscle architecture, the angle tool was used to measure the fascicle angle by tracing a clearly visible muscle fascicle the distance between the superficial and the deep aponeurosis. Fascicle length was measured using the straight-line tool, which was carefully placed on top of the previously described fascicle angle line to quantify the length of the fascicle of interest.

### 2.7. Maximal Muscle Strength

Isometric strength assessments were performed on a calibrated isokinetic dynamometer (HUMAC Norm, Computer Sports Medicine Inc., Stoughton, MA, USA). Participants were seated and secured into the dynamometer with straps placed over the participants’ chest, pelvis, and thigh in accordance with the manufacturer’s guidelines. Testing was performed on the participants’ dominant leg; leg dominance was determined according to the Harris Test of Lateral Dominance, where participants were asked which leg they would use to kick a soccer ball [26]. Prior to maximal strength testing, participants performed a warm-up consisting of three submaximal isometric voluntary muscle actions at 50%, 50%, and 75% of their perceived maximum for 3 to 5 s. Participants then performed 3 maximal voluntary contractions (MVCs) held for 3 to 5 s with 2 min of rest between repetitions. During each MVC, participants received strong verbal encouragement in which they were instructed to “kick out as hard and as fast as possible” [27]. All maximal testing occurred at a knee joint angle of 60° below the horizontal plane.

### 2.8. Signal Processing

All torque data were recorded at a sampling rate of 1926 Hz (Delsys, Inc., Natick, MA, USA) and stored for offline analysis. The torque signals were processed in a custom written software program (LabVIEW 2023, National Instruments, Austin, TX, USA). Torque signals were filtered using a recursive, second-order, zero-phase sift low-pass Butterworth filter. Isometric MVC peak torque (PT) was determined at the highest 500 ms epoch during the entire 3–4 s MVC plateau. Rapid torque (RT) was quantified from the linear slope of the torque-time curve at intervals of 25 ms (RT25), 50 ms (RT50), 100 ms (RT100), 200 ms (RT200), and 250 ms (RT250) following the onset of the MVC.

### 2.9. Eccentric Muscle Damage Exercise Protocol

The EMD exercise protocol occurred via the aforementioned isokinetic dynamometer. The range of motion for the exercise intervention was from 0° to 90° (full extension). Each participant performed ten sets of 30 repetitions for maximal and consecutive eccentric leg extension muscle actions at 45°/second. Between each muscle action, the leg passively returned to the starting position of the eccentric muscle action. Participants were given a minute and a half of rest between each set. For all repetitions, participants were instructed to “kick out” as hard as possible against the lever-arm during each of the aforementioned eccentric muscle actions. Strong verbal encouragement was supplied by the research team during the EMD exercise protocol for all participants.

### 2.10. Vertical Jump Assessment

To determine vertical jump velocity and power, participants performed three maximal effort countermovement vertical jumps using a linear position transducer (Tendo Sports Machines, Trencin, Slovakia). The linear position transducer was positioned on the floor behind the participant and was attached to a belt on the participant’s waist during the vertical jump assessment. The participant was positioned to allow the cord to be vertically extended without impeding vertical jump technique in accordance with the user’s guide (Tendo Weightlifting Analyzer, Microcomputer User’s Manual, Trencin, Slovak Republic). Participants were instructed to assume an akimbo stance (i.e., hands on hips, elbows bent outwards) for each jump. The highest vertical jump velocity and power of the three trials was used for analyses.

### 2.11. Subjective Ratings of Soreness Assessment

Prior to each of the testing sessions (i.e., pre, post24, post48, and post72) and following the eccentric muscle-damaging protocol (post0), the subjects completed a VAS to indicate their current level of soreness. Participants were asked about soreness at rest, during palpation, and during movement involving the targeted musculature (i.e., bodyweight squat). The same investigator palpated each of the following sites along the VL muscle belly: 5 cm proximal to the muscle midpoint, the muscle midpoint, and 5 cm distal to the muscle midpoint. Participants provided ratings on a 100 mm scale that ranged from “not sore” (0 mm) to “very, very sore” (100 mm).

### 2.12. Statistical Analysis

Primary outcomes included: (1) RT produced at 25, 50, 100, 200, and 250 ms following the onset of the muscular contraction during dynamometry testing, as well as the PT over a 500 ms epoch; (2) muscular characteristics (CSA and echo intensity) from ultrasonography along the length of the muscle; and (3) subjective muscle soreness and pain ratings, as established by a VAS. Secondary outcomes included: (1) thigh circumference and leg volume from 3DO; and (2) vertical jump power and velocity from a linear force transducer.

Each outcome was examined using a linear mixed-effects model with a random intercept for participant, a random slope, and a 1st order autocorrelation structure. For most outcomes, the random slope was structured as the time nested within participants, although a random slope for participants was used for VAS outcomes to allow for model convergence. A random intercept was selected to allow for the differing baseline quantities of each outcome, and a random slope was selected to allow for differing changes over the study period. The 1st order autocorrelation structure was selected to allow for the modeling of temporal dependencies in longitudinal data and to account for the correlation between repeated measures within the same participant. Models were fit by maximizing the restricted log-likelihood (REML). For each model, the fixed effects of conditions (CON and sMLCT) and time (pre, post0, post24, post48, and post72) were examined, along with their interaction. Model fit was quantified by the marginal R^2^ and conditional R^2^, indicating the proportion of variance explained by fixed effects only and fixed plus random effects, respectively.

Data were examined for extreme outliers (i.e., above or below 3X the interquartile range). With the exception of vertical jump peak power, no extreme outliers were found in the present dataset. Sensitivity analysis was performed and indicated that the presence of the one extreme outlier for vertical jump peak power did not meaningfully influence results, so this individual was retained in the analysis. As described above, VAS outcomes were rank-transformed due to extreme outliers and other violations of model assumptions when using raw data, especially the normality of residuals. However, the raw data are displayed to enhance interpretability. For other outcomes, the approximate normality of model residuals was confirmed through the visual inspection of quantile–quantile plots. Homoscedasticity for fitted models was confirmed through the visual inspection of residuals vs. fitted plots.

Joint tests were performed to evaluate the statistical significance of multiple terms simultaneously. Additionally, the individual model coefficients and their significance are presented. To help control the familywise error rate for joint tests, the Benjamini–Hochberg [28] correction was applied to all *p*-values within each group of outcome variables, and statistical significance was accepted at *p* ≤ 0.05 for corrected *p*-values. Participant characteristics were examined using independent samples *t*-tests or chi-squared tests, as appropriate. Statistical analysis was performed using R software (v. 4.4.0) [29] with the nlme (v. 3.1-164) [30], emmeans (v. 1.10.1) [31], and sjPlot (v. 2.8.15) [32] packages.

## 3. Results

### 3.1. Participant Characteristics

Forty participants completed the study and were included in the present analysis (Table 1). In the entire sample, CON, and sMLCT, the races and ethnicities present were non-Hispanic Caucasian (*n* = 17, 10, 7, respectively), Asian (*n* = 11, 3, 8), Hispanic (*n* = 10, 6, 4), and Black (*n* = 2, 1, 1). Thirty-four participants (*n* = 17 in each group) reported natural menstruation, while six (*n* = 3 in each group) reported oral contraceptive use. Thirty-six participants (*n* = 17 and 19 in CON and sMLCT, respectively) reported right leg dominance, while four (*n* = 3 and 1 in CON and sMLCT, respectively) reported left leg dominance. There were no differences between the CON and sMLCT groups for the proportions of different races and ethnicities (*p* = 0.36), menstruation status (*p* = 1.0), or leg dominance (*p* = 0.60), as indicated by chi-squared tests.

There were no significant differences between conditions for typical exercise habits (Table 2), nor nutritional intake during the four-day period surrounding the exercise session (i.e., the 24 h prior to the muscle-damaging exercise bout through the 72 h post-exercise monitoring period).

### 3.2. Strength Recovery

There was a significant condition × time interaction effect (*F* = 3.08; *p* = 0.05) present for isometric PT; however, mixed model coefficients did not reveal statistically significant interactions between conditions and specific time points (*p* > 0.05) (Figure 1). While there was no significant effect of the conditions (*p* > 0.05) on isometric PT, there was for time (*F* = 15.40; *p* < 0.001). Follow-up analysis indicated that PT was significantly reduced at post0 (B = −40.77; t = −5.18; *p* < 0.001) and post24 (B = −22.82; t = −2.16; *p* = 0.03), but not at the other timepoints (*p* > 0.05) (Figure 2). For the RT variables, there were no interactions or main effects for conditions present for RT25, RT50, RT100, RT200, or RT250 (*p* > 0.05) (Figure 2). While there was not a main effect for time for RT25 (*p* > 0.05), there was for RT50, RT100, RT200, and RT250 (*F* = 7.11–13.69; *p* < 0.001) (Table 3). Follow-up procedures indicated that RT50 was significantly lower at post0 (B = −26.57; t = −2.92; *p* < 0.001) and post48 (B = −24.50; t = −2.06; *p* = 0.04) compared to pre, but no other time points were statistically different (*p* > 0.05). When compared to pre, RT100 was significantly reduced at post0 (B = −40.22; t = −3.88; *p* < 0.001), post24 (B = −29.21; t = −3.14; *p* = 0.002), post48 (B = −42.10; t = −3.66; *p* < 0.001), and post72 (B = −37.43; t = −3.66; *p* < 0.001). Similarly, RT200 was significantly reduced at post0 (B = −55.02; t = −5.28; *p* < 0.001), post24 (B = −33.02; t = −3.49; *p* < 0.001), post48 (B = −40.48; t = −3.80; *p* < 0.001), and post72 (B = −22.18; t = −2.00; *p* = 0.04). Finally, RT250 was significantly reduced at post0 (B = −55.63; t = −5.51; *p* < 0.001), post24 (B = −32.75; t = −3.46; *p* < 0.001), post48 (B = −37.58; t = −3.71; *p* < 0.001), but not post72 (*p* > 0.05). Marginal R^2^ values for the strength models ranged from 0.05 to 0.11, with conditional R^2^ values of 0.86 to 0.94. Time-dependent reductions in PT and RT measures occurred post-intervention, with no main condition effect, but significant decreases were observed in several RT metrics throughout the follow-up visits (i.e., post0–post72).

### 3.3. Muscle Characteristics

There were no statistically significant interactions or main effects for VL CSA at any measurement location (*p* > 0.05). Regarding the echo intensity, time main effects were present at 50% (*F* = 5.33, *p* < 0.001) and 75% (*F* = 3.91, *p* = 0.04) of the muscle length (Figure 3 and Figure 4; Table 2). Following Benjamini–Hochberg correction procedures, at 50% muscle length, the post0 echo intensity was significantly greater (B = 5.32; t = 2.67, *p* = 0.008) than pre, but no other time points were altered. Similarly, at 75% of the muscle length, the post0 echo intensity was significantly greater (B = 4.55; t = 2.38, *p* = 0.02) than pre, but no other time points were found to be significantly different.

A significant mixed model coefficient indicated a lower echo intensity at 25% of the muscle length for the post-exercise (post0) time point in sMLCT. Marginal R^2^ values for fitted models ranged from 0.01 to 0.08, with conditional R^2^ values of 0.96 for all models. These results suggested that muscle edema was present immediately following the EMD protocol (i.e., post0) at 50 and 75% of the muscle length.

### 3.4. Subjective Ratings

When evaluated by a VAS while at rest and during a squat, there were no interactions or condition effects for pain (*p* > 0.05); however, there was a main effect for time when resting (*F* = 36.91; *p* < 0.001) and during a squat (*F* = 28.70; *p* < 0.001). Follow-up analysis indicated that pain was higher, relative to pre, at post0 (B = 1.84; t = 7.06; *p* < 0.001), post24 (B = 0.87; t = 3.44; *p* < 0.001), and post48 (B = 0.70; t = 2.77; *p* < 0.001). No significant differences for pain occurred at post72 (*p* > 0.05). During the bodyweight squat, follow-up analysis indicated that pain was similarly elevated at post0 (B = 2.17; t = 6.56; *p* < 0.001), post24 (B = 1.62; t = 5.73; *p* < 0.001), and post48 (B = 1.26; t = 4.26; *p* < 0.001). There was no significant difference in pain during the body weight squat at the post72 time point (*p* > 0.05).

Subjective pain was also assessed along three sites (proximal, middle, distal) on the thigh. There were no interactions or main effects for time or the conditions present for pain examined at the proximal location (Figure 5 and Figure 6; Table 2). Mixed model coefficients indicated a significantly lower pain rank for the proximal thigh in sMLCT at 72 h post-exercise (Figure 6). While mixed models were performed using rank-transformed data, raw data are presented to aid interpretability (Figure 5). Marginal and conditional R^2^ values for these models both ranged from 0.07 to 0.45.

Similarly, there were no interactions or main effects for the conditions present for pain at the middle or distal locations; however, there was a significant main effect for time at the middle (*F* 7.01; *p* < 0.001) and distal (*F* = 8.70; *p* < 0.001) locations. Follow-up analysis within the middle location indicated that, relative to pre, pain was significantly elevated at post0 (B = 1.16; t = 3.80; *p* < 0.001), post24 (B = 1.01; t = 3.65; *p* < 0.001), and post48 (B = 0.68; t = 2.41; *p* = 0.02). There were no significant differences for middle location pain at post72 (*p* > 0.05). Similarly, pain assessed distally, relative to pre, was elevated at post0 (B = 1.24; t = 4.07; *p* < 0.001), post24 (B = 0.98; t = 3.54; *p* < 0.001), post48 (B = 0.69; t = 2.46; *p* = 0.01), and post72 (B = 0.79; t = 2.81; *p* = 0.005). The results from the subjective pain rating suggested that the EMD protocol effectively induced muscle damage, resulting in significant increases in pain levels from baseline up to the 48 h time point.

### 3.5. Thigh Measurements

For the thigh circumference and leg volume measurements, there were no significant condition × time interactions or main effects for the conditions (F = 0.40–1.44, *p* = 0.36–0.81). Though, after collapsing across conditions, there were significant main effects for time (F = 5.47, *p* < 0.001) (Figure 7 and Figure 8; Table 2). Marginal R^2^ values for these models ranged from 0.04 to 0.07, and conditional R^2^ values ranged from 0.97 to 0.99. While condition had no impact, both thigh circumference and leg volume changes over time suggested that inflammation may have occurred following EMD.

### 3.6. Vertical Jump 

There were no interactions or main effects for the conditions (*p* > 0.05) when examining the vertical jump power or velocity. However, there were main effects for time for the average vertical jump power (*F* = 8.15, *p* <0.001) and velocity (*F* = 7.71, *p* < 0.001) metrics (Table 2; Figure 9 and Figure 10). Mixed model coefficients indicated significantly higher average power and average velocity in sMLCT at the 24 h post-exercise time point, as well as a significantly higher peak velocity at 48 h post-exercise (Figure 10). Marginal R^2^ values for these models ranged from 0.04 to 0.09, and conditional R^2^ values were 0.95 for all models.

Furthermore, when collapsed across the conditions and compared to pre, there was a decline in the average vertical jump power at post0 (B = −22.57; t = −1.95; *p* = 0.05), but power returned to pre-exercise values at all other timepoints (*p* > 0.05). Regarding the average velocity, after collapsing across the conditions and compared to pre, there was a similar decline in the velocity at post0 (B = −0.03; t = −2.05; *p* = 0.04), but not at any other time points (*p* > 0.05). Although the conditions had no main effects on the vertical jump metrics, significant time effects indicated short-term reductions in the power and velocity post-exercise, with sMLCT showing minor improvement at post24 and post48.

## 4. Discussion

### 4.1. Primary Findings

Amid growing interest surrounding effective nutritional aids for recovery, structured triglycerides have emerged as a potential dietary approach to support tissue repair following muscle damage [7,8]. Previous works have investigated related compounds’ influence on aerobic performance [8], but limited works have investigated their effect on recovery from exercise-induced muscle damage. The purpose of this study was to establish the effect of a novel formulation of sMLCT on muscle performance recovery following an EMD exercise protocol. While the influence of sMLCT on muscle function is still unclear, to our knowledge, this is the first study to examine the effect of sMLCT on muscle recovery following an EMD exercise protocol. The primary findings of the current study indicated that the short-term ingestion of sMLCT did not significantly influence the rate of muscle strength recovery following 300 maximal eccentric muscle actions, as compared to CON.

Previous works have demonstrated that sMLCT have had favorable impacts on tissue repair in patients with digestive disorders [11,12,13] and those in critical care settings [14,15]. Two meta-analyses suggest that sMLCT improves nutritional efficacy by enhancing nitrogen balance and increasing prealbumin levels, which support improved protein levels and increased nutrient utilization during acute care [14,15]. The nutrients from sMLCT alone are sufficient to maintain energy balance in patients receiving parenteral nutrition. During parenteral nutrition, the liver is a critical factor in the administration of sMLCT, functioning as a central determinant of protein synthesis and an overall indicator of metabolic response [14,15]. The improvement of nutritional efficacy in critical care settings may be related to the immune-modulating properties of saturated fatty acids and triglycerides, including their roles in cell membrane function and inflammatory response [9]. However, these data should be interpreted with caution. While they provide mechanistic insights into the potential action of sMLCTs, the findings may not be generalizable outside of critical care patients or those on parenteral nutrition. With that said, despite the mechanistic evidence available, after undergoing EMD in the present study, there were no significant differences in the response between the sMLCT and the CON groups following the acute exercise bout. Specifically, there were no condition × time interaction effects present for most performance variables; however, time main effects were present on select performance variables (e.g., vertical jump, PT, rapid strength), potentially indicating the presence of muscle damage. The lack of a significant interaction effect may be due to a number of factors including the degree of baseline inflammation present in the sample in addition to the dosage and characteristics of the dietary supplement. There are distinct differences between the available literature and that of the current study, particularly that the current participants were healthy college-aged females, as opposed to those undergoing parenteral nutrition in a critical care setting. Indeed, this study provides critical insight into the potential efficacy of sMLCT on muscle recovery in a healthy female population, while also highlighting the need for additional population-specific investigations.

### 4.2. Muscular Characteristics

In this investigation, 3DO and B-mode ultrasonography were used to quantify the inflammatory response following the EMD protocol on the whole-body and skeletal muscle levels. While previous studies [33,34] have reported sMLCT to provide a positive effect on skeletal muscle properties (i.e., muscle size) during energy expenditure, the present study did not indicate a significant change in skeletal muscle properties for any of the time points between groups. Likewise, 3DO-derived leg volume did not indicate a significant interaction effect. However, when collapsed across conditions, there was a significant main effect for time, similar to previous studies on eccentric exercise bouts [35,36]. This could potentially be due to the inflammatory process triggered by muscle damage [18,35,37]. Given that tissue and fluid can have an attenuation effect on ultrasound signals, previous works have utilized ultrasound echo intensity as a non-invasive marker of muscle damage [22,38,39]. In the present study, there was a significant increase in ultrasound echo intensity at the post-exercise (i.e., post0) time point, which could indicate that muscle edema was present within the VL [40,41]; however, this metric returned to normal upon follow-up testing. Collectively, these data suggest that while changes in muscle performance were noted, the acute effects of eccentric exercise were not present on the whole-body or skeletal muscle level.

### 4.3. Maximal and Rapid Strength

Within the current study, 10 sets of 30 repetitions were employed to induce muscle damage of the leg extensors and establish the effect of sMLCT on muscle recovery. Previous studies utilized PT and RT to examine the influence of exercise-induced muscle damage on recovery [18,42,43,44]. As recommended in previous research, PT and RT were also used as the metrics of muscle damage in the present study [19,20,21]. While there were no significant main effects for the conditions, PT was significantly reduced at the post0 and post24 time points, which was similar to a previous investigation [43]. Specifically, Hicks et al. [43] assessed EMD in the VL following six sets of 12 eccentric knee extensions and reported that significant reductions in PT were present one hour and forty-eight hours post exercise. The reductions in PT and RT observed in the current investigation further support the effectiveness of the EMD protocol at inducing quadricep muscle damage.

In the current investigation, there were main effects present during the later phases of RT (i.e., RT100 to RT250), but there were no main effects in the early phases (i.e., RT25). Similarly, previous studies [19,45] that utilized RT as an indicator of muscle damage indicated significant decreases in late-phase RT as opposed to early phase. The reductions in RT during the later phases may be due to different neuromuscular factors involved in torque generation [46]. When assessing the rate of torque development, neural activation and the motor unit discharge rate seem to contribute more in the early phase of contraction [21,46], while the late-phase rate of torque development is associated with maximal strength and muscle mass [46,47]. Moreover, the decreases in PT and RT in the present investigation may partly be due to the repeated bout effect [17,23,39,48,49,50]. The repeated bout effect refers to the neuromuscular response that occurs during the initial session of an exercise protocol (e.g., EMD protocol), with subsequent sessions of the same exercise resulting in less pronounced changes in muscular strength [23,50].

Earlier research has explored exercise-induced muscle damage and the mechanisms behind the repeated bout effect that may influence the neuromuscular response following an initial bout of exercise-induced muscle damage [17,23,39,48,49,50]. Previous studies [48,49] have also reported torque values returning close to baseline within 48 h following muscle damage induced by eccentric and concentric movements, likely due to the repeated bout effect. Increased neural activity and improvements in motor unit recruitment may account for the lack of differences in torque at 48 h post-exercise [48,49]. Furthermore, Molina and Denadai [17] implemented a similar damaging protocol to the one used in the current investigation and observed reductions in PT values following the initial bout of muscle damage, much like that seen in the present work. In the current investigation, the repeated bout effect may have occurred, resulting in reduced PT and RT at the initial time points (i.e., post0 and post24), with recovery by the subsequent time points (i.e., post48 and post72). Early reductions in PT and RT following muscle damage may be attributed to neural response and a broader workload distributed across active muscle fibers and the magnitude of muscle lengthening [50].

### 4.4. Vertical Jump Performance

Vertical jump performance provides insight into muscle function and neuromuscular adaptations that may occur following an exercise intervention (e.g., resistance training) [16,51,52]. In the present study, there was no main effects for the conditions on the vertical jump power or velocity following the acute EMD protocol. However, there was a main effect for time, which indicated that vertical jump performance (i.e., velocity and power) at the pre and post-exercise time points decreased. The main effect observed in the current study was similar to previous work [16,53]. Byrne and Eston [16] evaluated vertical jump performance using three different jump types (squat, drop, and countermovement jump) and found significant reductions from pre- to post-exercise following a 100 barbell squat muscle-damaging protocol. Notably, in the current study, there were significant main effects for time and jump performance from pre- to immediate post-muscle damage. While speculative, these outcomes might indicate that damaged muscle impacts the utilization of the stretch shortening cycle, which is known to store and transform elastic energy within the muscle [16,53,54].

### 4.5. Subjective Ratings of Soreness

The subjective severity of DOMS was completed on a VAS to evaluate muscle soreness at rest, during movement, and while palpated. Previous studies have used the measures of DOMS (e.g., swelling, inflammation, and muscle fiber damage) to assess muscle soreness following a bout of exercise [19,23,35,36,37,55,56,57]. The present work reported a main effect for time for each VAS assessment, suggesting that subjective discomfort was higher at post0, post24, and post48, compared to pre. While the present study’s absolute reports of discomfort are low, these findings were similar to previous investigations [57]. Peak DOMS is frequently observed 24 and 48 h after eccentric exercise [35,36], while muscle regeneration and repair occurs around 72 h post-exercise [37]. These findings aligned with the current study, where self-reported soreness was prominent up to the 48 h time point but subsided by the final timepoint (i.e., 72 h post-exercise). In related studies [23,35,55] utilizing multiple time points to assess DOMS following damaging exercise, the ratings of perceived muscle soreness returned to baseline by the 72 h time point. Notably, a main effect for VAS-reported distal pain of the VL was observed at all post-EMD time points (i.e., post0, post24, post48, and post72). Observed soreness at the distal portion of the muscle was in agreement with previous studies, which suggests that the myotendinous junctions may be the primary site for muscle soreness to occur [58,59,60]. Moreover, distal soreness in the quadriceps could be more pronounced due to the transition in fiber type near the musculotendinous junction and the mechanical demand of the distal muscle fibers during a knee extension [60].

### 4.6. Study Strengths and Limitations

It is important to discuss the limitations of the present study. The current sample only included recreationally active females; thus, it is unclear how sMLCT would influence muscle performance recovery after EMD in a population consisting of males or those with different training statuses (e.g., athletes, sedentary, etc.). Additionally, muscle damage was induced using unilateral movement of the dominant limb as opposed to bilateral movements. Future studies may consider investigating how different movement patterns (e.g., bilateral, ballistic) may influence performance and recovery when supplementing with sMLCT. Further, the current investigation did not assess creatine kinase or other relevant biomarkers. In a previous study [39], plasma creatine kinase was used as a biomarker to assess muscle damage following an EMD protocol and significant differences were observed at follow-up days 3–5. As sMLCT may influence immunomodulatory and metabolic processes, future works may assess cytokine profiles or the markers of oxidative stress to provide additional insights on the potential discrepancy between any noted physiological and performance metrics. Future studies should evaluate inflammatory biomarkers along with objective and subjective performance metrics to obtain more comprehensive findings surrounding the influence of sMLCT on muscle recovery. Lastly, it is possible that the dosage or duration of the supplementation protocol may have been insufficient to aid in the repair and recovery of skeletal muscle tissue following a bout of eccentric-only muscle damage.

## 5. Conclusions

The findings from this double-blind, randomized, placebo-controlled study reveal that—within the context of this study—sMLCT supplementation does not appreciably affect muscle performance or recovery following an acute EMD protocol. While time-related improvements were observed in recovery metrics (i.e., inflammatory response, PT, RT, vertical jump, and DOMS), these changes were independent of the supplementation conditions (i.e., sMLCT and CON). The lack of a direct effect based on the condition (i.e., sMLCT, CON) suggests that short-term sMLCT supplementation is unlikely to enhance recovery in recreationally active females following an EMD protocol, within the context of the investigated supplementation and exercise protocol. Further research is needed to determine the potential role of sMLCT in enhancing muscular performance and recuperation strategies under different conditions. Until further evidence emerges, clinicians and practitioners may utilize caution when considering the use of short-term sMLCT supplementation in efforts to enhance muscle recovery in recreationally active females.

## Figures and Tables

**Figure 1 nutrients-17-01604-f001:**
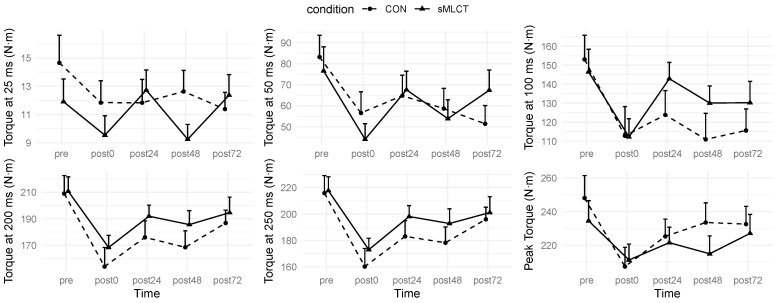
Muscular strength changes. Data displayed as mean ± SE.

**Figure 2 nutrients-17-01604-f002:**
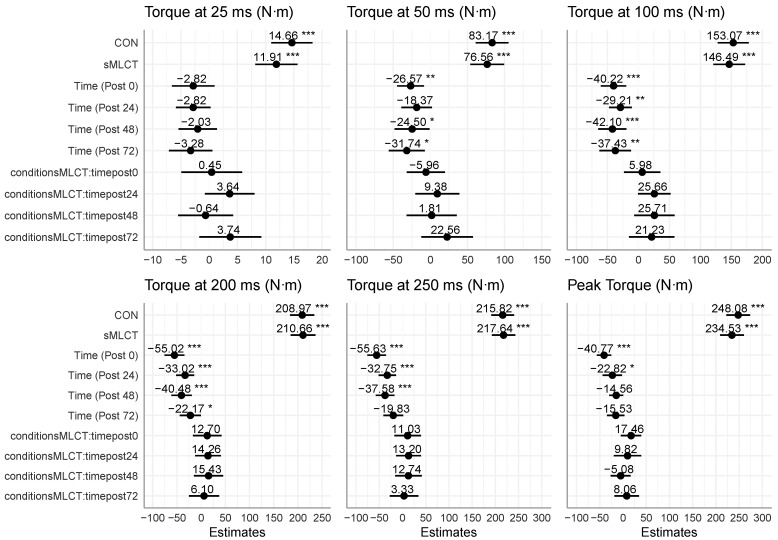
Mixed model coefficients for strength models. * indicates *p* < 0.05; ** indicates *p* < 0.01; *** indicates *p* < 0.001.

**Figure 3 nutrients-17-01604-f003:**
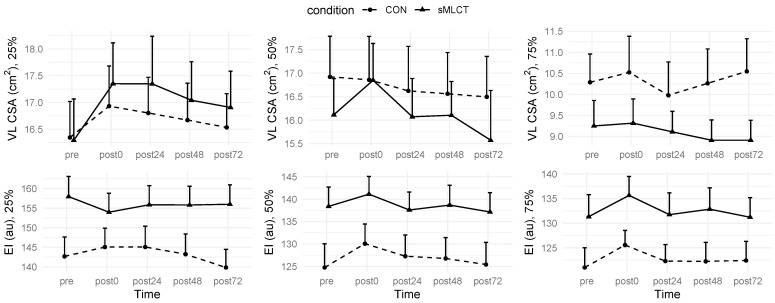
Changes in muscle characteristics. Data displayed as mean ± SE.

**Figure 4 nutrients-17-01604-f004:**
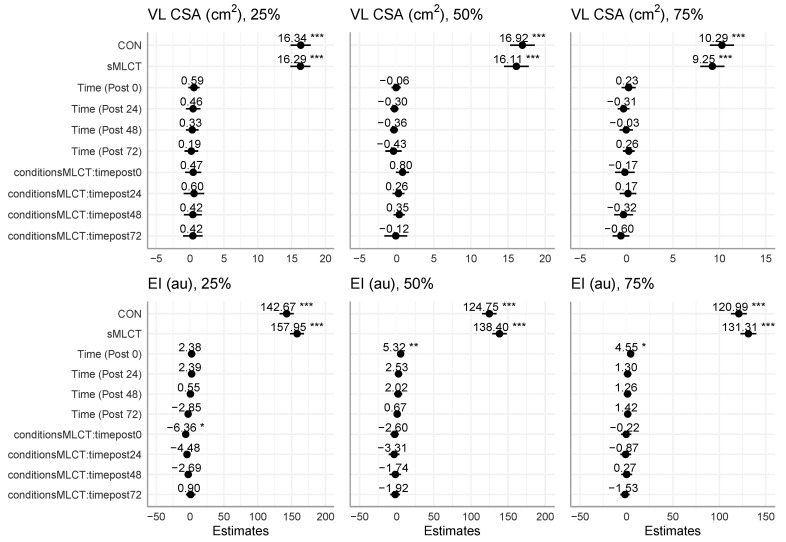
Mixed model coefficients for muscle characteristic models. * indicates *p* < 0.05; ** indicates *p* < 0.01; *** indicates *p* < 0.001.

**Figure 5 nutrients-17-01604-f005:**
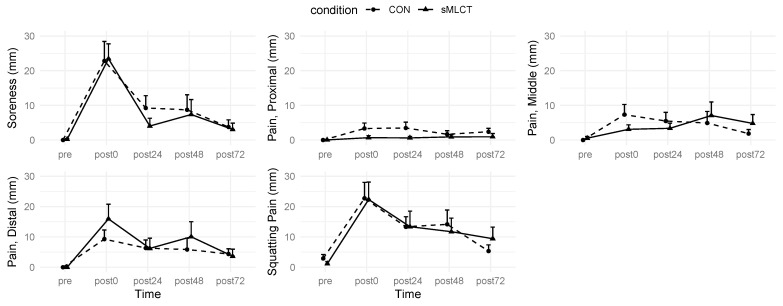
Changes in Subjective Variables. Data displayed as mean ± SE.

**Figure 6 nutrients-17-01604-f006:**
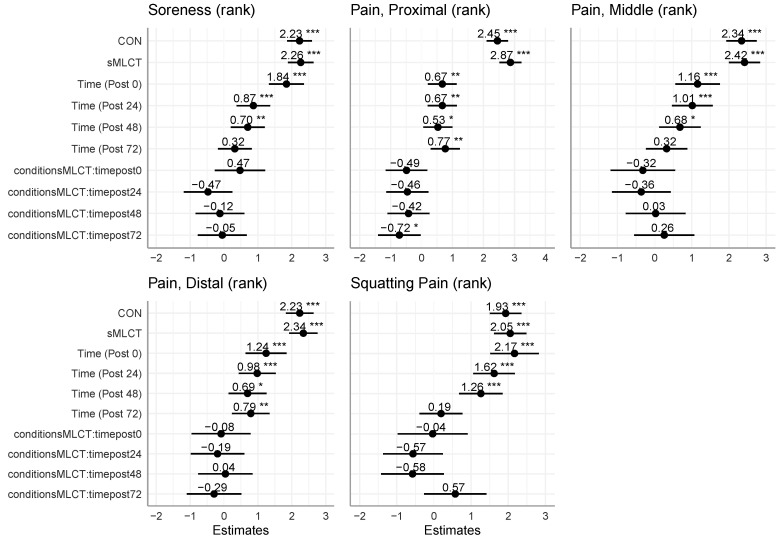
Mixed model coefficients for subjective variables. * indicates *p* < 0.05; ** indicates *p* < 0.01; *** indicates *p* < 0.001.

**Figure 7 nutrients-17-01604-f007:**
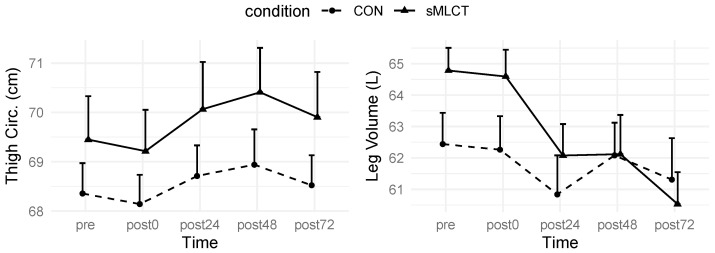
Changes in thigh anthropometry.

**Figure 8 nutrients-17-01604-f008:**
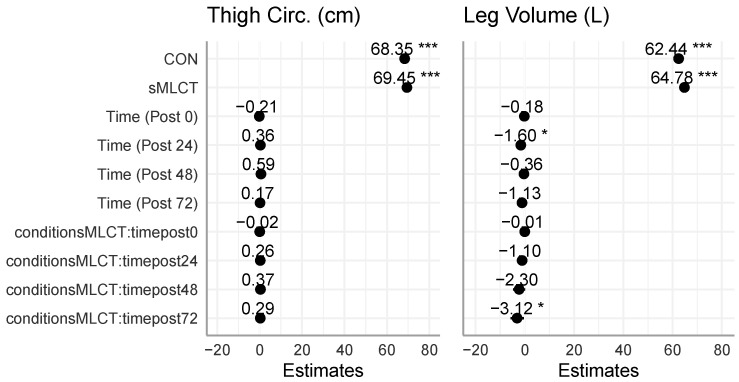
Mixed model coefficients for thigh anthropometry. * indicates *p* < 0.05; *** indicates *p* < 0.001.

**Figure 9 nutrients-17-01604-f009:**
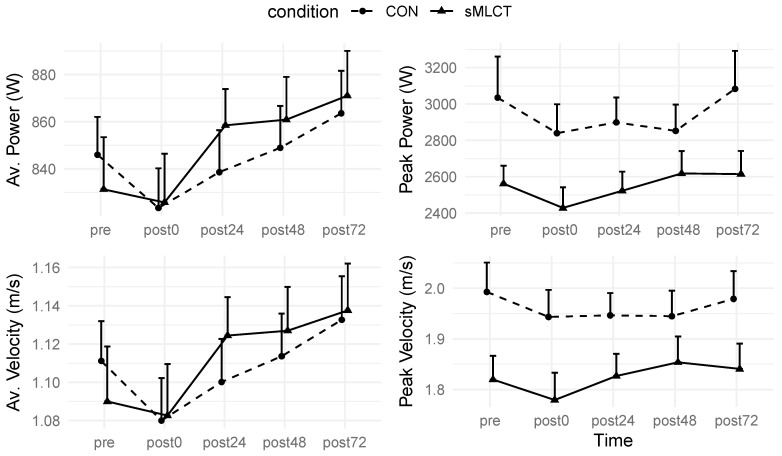
Changes in vertical jump performance.

**Figure 10 nutrients-17-01604-f010:**
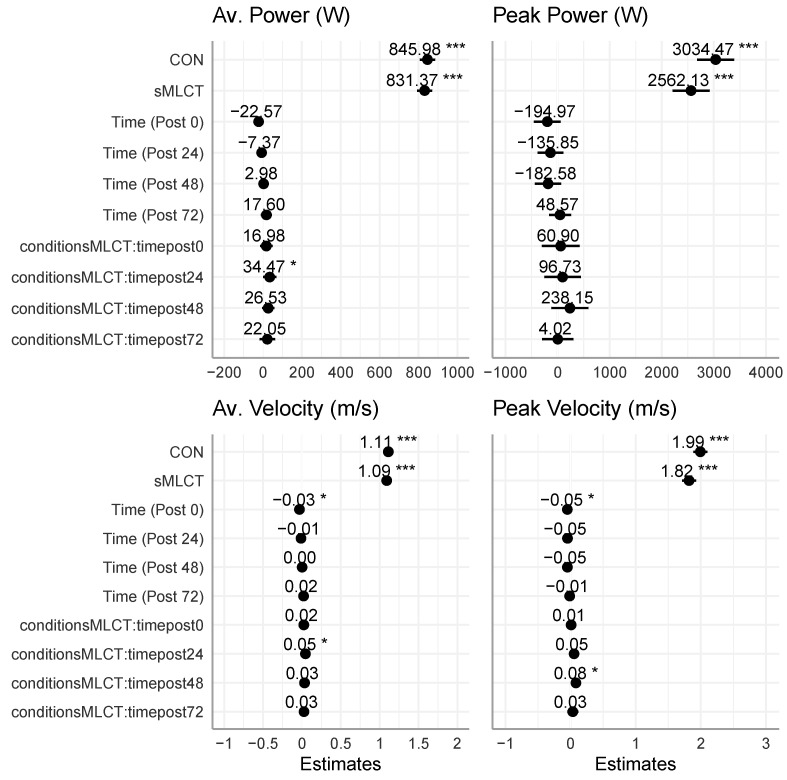
Mixed model coefficients for vertical jump performance. * indicates *p* < 0.05; *** indicates *p* < 0.001.

**Table 1 nutrients-17-01604-t001:** Participant characteristics.

	All (*n* = 40)		CON (*n* = 20)		sMLCT (*n* = 20)		
	Mean	SD	Mean	SD	Mean	SD	*p* ^1^
Age (year)	22.3	3.4	22.3	3.4	22.2	3.5	0.93
Height (cm)	162.0	5.4	161.9	4.9	162.1	6.0	0.93
Weight (kg)	61.7	9.3	60.3	10.3	63.1	8.3	0.84
BMI (kg/m^2^)	23.5	3.4	23.0	3.5	24.1	3.3	0.84
BF%	33.3	7.1	30.8	6.8	35.9	6.6	0.14
FFMI (kg/m^2^)	15.1	1.6	15.1	1.9	15.2	1.3	0.93
WHR	0.7	0.1	0.7	0.0	0.7	0.1	0.93

^1^ FDR-corrected *p*-values from independent sample *t*-tests. BMI: body mass index; BF%: body fat percentage; FFMI: fat-free mass index; WHR: waist-to-hip ratio.

**Table 2 nutrients-17-01604-t002:** Exercise habits and nutritional intake.

	All (*n* = 40)		CON (*n* = 20)		sMLCT (*n* = 20)		
	Mean	SD	Mean	SD	Mean	SD	*p* ^1^
*Exercise Habits*							
Exercise Frequency (days/week)	1.7	1.2	1.8	1.0	1.6	1.4	0.69
Years of Exercise	1.5	2.0	2.2	2.5	0.9	1.0	0.10
RT Frequency (days/week)	0.1	0.2	0.1	0.3	0.0	0.0	0.24
RT Duration (min/session)	1.2	5.3	2.4	7.3	0.0	0.0	0.24
ET Frequency (days/week)	1.0	1.4	1.0	0.9	1.0	1.7	1.00
ET Duration (min/session)	15.9	17.8	20.1	19.2	11.6	15.7	0.24
CT Frequency (days/week)	0.1	0.4	0.2	0.5	0.0	0.0	0.24
CT Duration (min/session)	2.3	10.4	4.6	14.5	0.0	0.0	0.24
Activity Expenditure (kcal/week)	2547.9	4376.9	3307.1	5793.1	1788.7	2133.0	0.32
*Nutritional Intake*							
Energy (kcal)	1431.0	416.4	1596.9	426.9	1265.1	340.1	0.08
Protein (g)	62.0	18.7	67.0	18.6	56.9	17.9	0.20
Fat (g)	62.7	20.0	70.8	18.8	54.7	18.1	0.08
Carbohydrate (g)	158.4	53.0	176.4	57.0	140.3	42.9	0.10
Water (g)	726.0	323.5	800.6	345.6	651.4	289.3	0.24
Saturated Fats (g)	22.0	8.0	25.1	8.7	18.9	6.1	0.08
Monounsaturated Fats (g)	20.7	7.0	23.1	6.5	18.3	6.9	0.10
Polyunsaturated Fats (g)	14.1	5.6	15.9	4.3	12.2	6.2	0.10

^1^ FDR-corrected *p*-values from independent sample *t*-tests. RT: resistance training; ET: endurance training; CT: concurrent training.

**Table 3 nutrients-17-01604-t003:** Joint tests for outcome variables.

Outcome	Term	F-Ratio	*p*
Torque at 25 ms (Nm)	condition	0.71	0.64
	time	1.37	0.56
	condition × time	2.88	0.06
Torque at 50 ms (Nm)	condition	0.01	0.92
	time	7.11	<0.001 *
	condition × time	1.12	0.63
Torque at 100 ms (Nm)	condition	0.42	0.67
	time	7.29	<0.001 *
	condition × time	1.25	0.59
Torque at 200 ms (Nm)	condition	0.65	0.64
	time	11.49	<0.001 *
	condition × time	0.49	0.78
Torque at 250 ms (Nm)	condition	0.51	0.66
	time	13.69	<0.001 *
	condition × time	0.51	0.78
Peak Torque (Nm)	condition	0.28	0.72
	time	15.40	<0.001 *
	condition × time	3.08	0.05
VL CSA at 25% (cm^2^)	condition	0.12	0.87
	time	2.50	0.19
	condition × time	0.24	0.91
VL CSA at 50% (cm^2^)	condition	0.22	0.82
	time	1.64	0.37
	condition × time	0.92	0.72
VL CSA at 75% (cm^2^)	condition	1.82	0.37
	time	0.82	0.72
	condition × time	1.05	0.69
EI at 25% (au)	condition	3.57	0.19
	time	0.81	0.72
	condition × time	2.46	0.19
EI at 50% (au)	condition	3.71	0.19
	time	5.33	<0.001 *
	condition × time	0.35	0.91
EI at 75% (au)	condition	3.40	0.19
	time	3.91	0.04 *
	condition × time	0.26	0.91
Soreness (rank)	condition	0.00	1.00
	time	36.91	<0.001 *
	condition × time	1.68	0.34
Pain, Proximal (rank)	condition	0.00	1.00
	time	2.30	0.15
	condition × time	1.13	0.64
Pain, Middle (rank)	condition	0.00	1.00
	time	7.01	<0.001 *
	condition × time	0.90	0.78
Pain, Distal (rank)	condition	0.00	1.00
	time	8.70	<0.001 *
	condition × time	0.20	1.00
Squatting Pain (rank)	condition	0.00	1.00
	time	28.70	<0.001 *
	condition × time	2.57	0.12
Thigh Circumference (cm)	condition	1.44	0.36
	time	5.47	<0.001 *
	condition × time	0.40	0.81
Leg Volume (L)	condition	0.63	0.52
	time	6.12	<0.001 *
	condition × time	1.59	0.36
Average VJ Power (W)	condition	0.05	0.88
	time	8.15	<0.001 *
	condition × time	1.14	0.42
Peak VJ Power (W)	condition	4.35	0.12
	time	1.55	0.33
	condition × time	1.12	0.42
Average VJ Velocity (m/s)	condition	0.03	0.88
	time	7.71	<0.001 *
	condition × time	1.28	0.42
Peak VJ Velocity (m/s)	condition	4.05	0.12
	time	2.52	0.12
	condition × time	1.68	0.32

* indicates *p* < 0.05.

## Data Availability

Data may be available from the corresponding author upon reasonable request and pending relevant approvals.

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
