# Peer review of "Influence of Structured Medium- and Long-Chain Triglycerides on Muscular Recovery Following Damaging Resistance Exercise"

_nutrients, 2025, doi:10.3390/nu17101604_

Round 1

Reviewer 1 Report

Comments and Suggestions for Authors

This study investigates the effect of structured medium and long chain triglycerides (sMLCT) on muscle performance and recovery following a muscle-damaging resistance exercise protocol. The authors hypothesise that sMLCT may serve as an effective delivery vehicle for medium-chain fatty acids to skeletal muscle, potentially enhancing recovery. Forty female participants were randomised into two groups (placebo and sMLCT) and underwent assessments of muscle performance, soreness and muscle size at various time points following exercise. The main findings suggest that sMLCT supplementation had no significant effect on the rate of muscle strength recovery following resistance exercise.

I have made some comments about how you could improve your work. This doesn't mean you have to agree or rewrite it the same way. It's just a suggestion and another way of looking at things to help you.

1) While the 'Introduction' section explains DOMS and its impact on exercise performance, it could benefit from a brief mention of why muscle recovery is essential for athletes, active individuals or those undergoing rehabilitation. Adding context on the wider importance of muscle recovery in different populations could better highlight the importance of this study. Example of my suggestion: "Efficient muscle recovery is critical not only for athletes, but also for individuals undergoing rehabilitation or regular physical activity to prevent injury and improve long-term performance."

2) Although the 'Introduction' section mentions the production of sMLCT and its effect on tissue repair, more detail on how sMLCT might specifically affect muscle recovery could be helpful. This could include discussion of the potential role of medium and long chain fatty acids in reducing inflammation or improving lipid metabolism, which could contribute to muscle repair. My suggestion: "Medium and long chain fatty acids have been shown to have anti-inflammatory properties which may help to reduce muscle inflammation after intense exercise, potentially speeding up the recovery process".

3) The 'Introduction' mentions the lack of research on sMLCT in muscle recovery, but it could provide more information on the existing research on other supplements aimed at improving muscle recovery (e.g. creatine, protein, omega-3). A brief mention of the results of these studies could emphasise why sMLCT might offer a novel approach or what differentiates it from other recovery aids. My suggestion: "While other supplements, such as creatine and protein, have been extensively studied for muscle recovery, the potential role of sMLCT in this area remains underexplored, providing an opportunity for innovation."

4) The ‘Introduction’ could emphasise more clearly why sMLCT is considered 'novel' in the context of this study. While the biochemical composition is mentioned, its novel application for muscle recovery is not fully explained. Clarifying the innovation behind the use of sMLCT specifically for recovery after muscle-damaging exercise would further clarify the relevance of the study.

5) The 'Introduction' could be tightened slightly to avoid repetition, particularly in the description of DOMS and its causes. The same points are repeated several times, which could be condensed to make the section more concise. My suggestion: "Delayed onset muscle soreness (DOMS), a common consequence of unaccustomed physical activity, typically occurs 24-48 hours after eccentric exercise and is primarily caused by muscle damage. DOMS causes discomfort and can impair muscle function, potentially affecting performance."

6) Although the study mentions that participants were randomised to either the placebo control group or the sMLCT group, it would be useful to specify how randomisation was performed. Was it computer generated and how were any potential biases in the randomisation process minimised? Providing this information can ensure the transparency and validity of the allocation process.

7) The study does a good job of explaining the approach used to control for the effects of the menstrual cycle by enrolling participants in the early follicular phase or, for those taking oral contraceptives, during the placebo pill week. It would be beneficial to add how this phase was confirmed (e.g. by cycle tracking or hormonal testing), as this could affect the reproducibility of the study.

8) The compliance monitoring through an online Survey Monkey questionnaire is mentioned, but further clarification could be provided regarding how the compliance with the dietary supplements was ensured. Were participants required to submit photographic evidence of their supplements, or were any other methods used to verify adherence?

9) The description of the ultrasound imaging and its analysis is quite thorough. However, it would be helpful to specify how many images were taken at each time point and the number of repetitions per scan. This would provide more detail on the consistency and reliability of the measurements taken.

10) The statistical analysis section is quite detailed, but it would be helpful to include more information on how the assumptions for the linear mixed effects models were tested.

11) Although the trial ensures that participants take their assigned treatment and follow exercise protocols, it would be useful to note if any additional environmental factors (e.g. room temperature, time of day, or dietary intake) were controlled during the trial. Environmental controls can sometimes influence physiological responses and should be documented.

12) The EMD exercise protocol is well described, but further information could be added on the intensity and rationale behind the choice of the specific range of motion (0° to 90°). Was there a specific reason for choosing this range of motion and how does it relate to typical exercise protocols for muscle damage?

13) In the 'Results' section, some results refer to "condition" in general, but it's not always clear whether CON vs. sMLCT results are being directly compared, or just discussed within a group. Add more between-group comparisons, especially for time points where differences are hypothesised or relevant to your aims.

14) Although the statistical details are rich, readers may miss the broader implications without interpretive summaries. End each subsection with 1-2 sentences summarising the main trend (e.g. "These results suggest that recovery was faster in sMLCT at early time points for RT100 and RT200, despite non-significant interaction effects.)

15) Tables 1 and 2 are mentioned, but basic group means and SDs for key variables (e.g. peak torque, RT metrics, pain scores) are not consistently reported in the text. Include brief summary statistics for important outcome variables for context (especially pre/post values where changes are discussed).

16) Some parts (e.g. repeated mention that post72 values weren't significant) could be condensed to avoid repetition. Group results for non-significant changes in a concise format (e.g. 'No significant differences were found at post72 for all RT variables except RT200').

17) There's a mix of reporting rank transformed data vs. raw data. Please clarify the rationale for this distinction (perhaps in the methods) or ensure consistent interpretation guidance.

18) While the study is presented as novel, the "so what" question is underdeveloped. Suggestion: Add a short paragraph at the end of 4.1 or the beginning of 4.6 to explicitly state the contribution of this research to the field of nutritional ergogenics and muscle recovery. My suggestion: "This study provides fundamental insight into the limited efficacy of sMLCT on muscle recovery in a healthy female population, highlighting the need for population-specific supplementation protocols".

19) The study focuses on performance metrics, but biochemical markers (e.g. CK, IL-6) were not measured. Suggestion: In 4.6 you mention this as a limitation, but it could also be used to more strongly frame a future research opportunity. My suggestion: "Given the proposed immunomodulatory and metabolic effects of sMLCT, future work incorporating cytokine profiles or oxidative stress markers could clarify whether there is a discrepancy between physiological processes and performance outcomes".

20) Much of the justification for sMLCT is based on studies in critical care patients or those on parenteral nutrition, which may not be generalisable. Consider condensing this discussion (lines 406-417) and emphasising why these findings may not apply to young, healthy individuals.

21) The interpretation of null results needs to be more nuanced. Rather than only stating "no significant effect," explore why the supplement might not have worked in this context. My suggestion: "The lack of significant condition × time interactions may be due to the relatively low baseline inflammation in the population studied, or to insufficient bioavailability or dosage of the active compounds."

22) There is little translation of the findings into practical advice for sports scientists or clinicians. Please add a concluding statement in 4.6 or a short new section (4.7 Practical implications). My suggestion: "Until further evidence emerges, practitioners may consider that short-term sMLCT supplementation is unlikely to improve recovery in recreational female athletes following eccentric muscle damage."

This manuscript presents valuable findings, but improvements in clarity, depth of discussion and methodological detail would further enhance its impact. I recommend acceptance of the manuscript with minor revisions.

Author Response

Reviewer 1

  • This study investigates the effect of structured medium and long chain triglycerides (sMLCT) on muscle performance and recovery following a muscle-damaging resistance exercise protocol. The authors hypothesise that sMLCT may serve as an effective delivery vehicle for medium-chain fatty acids to skeletal muscle, potentially enhancing recovery. Forty female participants were randomised into two groups (placebo and sMLCT) and underwent assessments of muscle performance, soreness and muscle size at various time points following exercise. The main findings suggest that sMLCT supplementation had no significant effect on the rate of muscle strength recovery following resistance exercise.

  • I have made some comments about how you could improve your work. This doesn't mean you have to agree or rewrite it the same way. It's just a suggestion and another way of looking at things to help you.

  • While the 'Introduction' section explains DOMS and its impact on exercise performance, it could benefit from a brief mention of why muscle recovery is essential for athletes, active individuals or those undergoing rehabilitation. Adding context on the wider importance of muscle recovery in different populations could better highlight the importance of this study. Example of my suggestion: "Efficient muscle recovery is critical not only for athletes, but also for individuals undergoing rehabilitation or regular physical activity to prevent injury and improve long-term performance."
    1. Thank you for this comment as we believe it is valuable. We have tried to implement these changes suggested by the reviewer within our introduction.
      1. “Efficient muscle recovery is critical for not only athletes, but for the general population. Specifically, individuals interested in preventing injury or whom are undergoing rehabilitation may be particularly interested in the role of muscle recovery on long-term performance outcomes [3].”

  • Although the 'Introduction' section mentions the production of sMLCT and its effect on tissue repair, more detail on how sMLCT might specifically affect muscle recovery could be helpful. This could include discussion of the potential role of medium and long chain fatty acids in reducing inflammation or improving lipid metabolism, which could contribute to muscle repair. My suggestion: "Medium and long chain fatty acids have been shown to have anti-inflammatory properties which may help to reduce muscle inflammation after intense exercise, potentially speeding up the recovery process".
    1. Thank you for this comment as we believe it is valuable, Reviewer 2 had a similar concern. In response, we have made an edit that bolsters a mechanistic insight of sMLCT in non-clinical populations. However, if either reviewer believes that additional discussion is required, we are happy to consider this on future iteration of review.
      1. “Although there is limited research on sMLCT in non-clinical populations, the physiological mechanism by which medium- and long-chain fatty acids aid muscle tissue recovery is by promoting a positive nitrogen balance [14, 15] and potential anti-inflammatory properties [15], therefore providing strong rational of the use of sMLCT in athletic settings.”

  • The 'Introduction' mentions the lack of research on sMLCT in muscle recovery, but it could provide more information on the existing research on other supplements aimed at improving muscle recovery (e.g. creatine, protein, omega-3). A brief mention of the results of these studies could emphasise why sMLCT might offer a novel approach or what differentiates it from other recovery aids. My suggestion: "While other supplements, such as creatine and protein, have been extensively studied for muscle recovery, the potential role of sMLCT in this area remains underexplored, providing an opportunity for innovation."
    1. We appreciate this comment. With additional changes made to the introduction, the authors believe that this comment may stray away from the purpose of the study. However, if the reviewer disagrees we are happy to reconsider this on subsequent rounds of review.

  • The ‘Introduction’ could emphasise more clearly why sMLCT is considered 'novel' in the context of this study. While the biochemical composition is mentioned, its novel application for muscle recovery is not fully explained. Clarifying the innovation behind the use of sMLCT specifically for recovery after muscle-damaging exercise would further clarify the relevance of the study.
    1. This is a very valuable comment, and we appreciate the feedback. We have tried to expand and add more clarity at the end of second paragraph of the introduction and believe this has adequately addressed your concern.
      1. “Although there is limited research on sMLCT in non-clinical populations, the physiological mechanism by which medium- and long-chain fatty acids aid muscle tissue recovery is by promoting a positive nitrogen balance [14, 15] and potential anti-inflammatory properties [15], therefore providing strong rational of the use of sMLCT in athletic settings.”

  • The 'Introduction' could be tightened slightly to avoid repetition, particularly in the description of DOMS and its causes. The same points are repeated several times, which could be condensed to make the section more concise. My suggestion: "Delayed onset muscle soreness (DOMS), a common consequence of unaccustomed physical activity, typically occurs 24-48 hours after eccentric exercise and is primarily caused by muscle damage. DOMS causes discomfort and can impair muscle function, potentially affecting performance."
    1. We thank the reviewer for their interest in maximizing the clarity of our work. However, we are unsure of exactly where our points are “repeated several times”. We did shorten paragraph 1, though if future edits are required, we request the reviewer specific our areas of overlap in the introduction so that we can accurately make the requested changes.

  • Although the study mentions that participants were randomised to either the placebo control group or the sMLCT group, it would be useful to specify how randomisation was performed. Was it computer generated and how were any potential biases in the randomisation process minimised? Providing this information can ensure the transparency and validity of the allocation process.
    1. Thank you for your astute observation. We have clarified this comment in the methods section, detailed that randomization occurred a priori with the use of an external software program.
      1. “Randomization procedures were performed a priori using R software (v. 4.4.0; randomizeR package) following enrollment. Randomization was stratified by age (18 – 24 vs. 25 – 35), body mass (50 – 80 kg vs. 80 – 110 kg), and hormonal birth control use (oral contraceptive users vs. naturally menstruating).”

  • The study does a good job of explaining the approach used to control for the effects of the menstrual cycle by enrolling participants in the early follicular phase or, for those taking oral contraceptives, during the placebo pill week. It would be beneficial to add how this phase was confirmed (e.g. by cycle tracking or hormonal testing), as this could affect the reproducibility of the study.
    1. Thank you for this comment as we believe it is valuable. We have attempted to add justification how menstruation was confirmed.
      1. “Menstruation was confirmed via oral interview upon visiting the laboratory during the familiarization visit.”

  • The compliance monitoring through an online Survey Monkey questionnaire is mentioned, but further clarification could be provided regarding how the compliance with the dietary supplements was ensured. Were participants required to submit photographic evidence of their supplements, or were any other methods used to verify adherence?
    1. We appreciate this valuable feedback on the supplement compliance. While our methods clearly describe our compliance monitoring procedures, we did not use additional compliances techniques (i.e., photographic evidence).

  • The description of the ultrasound imaging and its analysis is quite thorough. However, it would be helpful to specify how many images were taken at each time point and the number of repetitions per scan. This would provide more detail on the consistency and reliability of the measurements taken. 2 images and best one was analyzed
    1. Your comment is well received. We have adjusted our manuscript to acknowledge these details regarding the methods for ultrasound assessment at the end of section 2.5
      1. “At each visit, two scans were taken at each scanning site, and the best image was used for analysis for that testing day.”

  • The statistical analysis section is quite detailed, but it would be helpful to include more information on how the assumptions for the linear mixed effects models were tested.
    1. This is an astute observation by the reviewer. Model diagnostics were completed via visual observation of the Q-Q plot and other approaches, as detailed in section 2.12. If additional clarification is requested, we are happy to revisit this item.

  • Although the trial ensures that participants take their assigned treatment and follow exercise protocols, it would be useful to note if any additional environmental factors (e.g. room temperature, time of day, or dietary intake) were controlled during the trial. Environmental controls can sometimes influence physiological responses and should be documented.
    1. Thank you for this observation. The authors have tried to address the controls of the environmental factors within section 2.3 Pre- and Post-visit procedures.

“For each of the testing visits (i.e., pre to post72), participants arrived at the laboratory at the same time of day (± ~2h) and consumed one of two daily doses of the assigned treatment. The laboratory was kept at a constant temperature (22 °C).”

  • The EMD exercise protocol is well described, but further information could be added on the intensity and rationale behind the choice of the specific range of motion (0° to 90°). Was there a specific reason for choosing this range of motion and how does it relate to typical exercise protocols for muscle damage?
    1. Within the muscle damage literature, the details of the EMD protocol are critical, and we appreciate this reviewer’s critical eye. When the authors designed the study, we compared what was commonly seen in the literature (a broad range of ROM’s) with the physiological mechanisms of EMD commonly seen with exercise training. Not only is it common for many exercises to use a full 90 degrees ROM, but it is also a way to experimentally ensure the most amount of physical work can be performed by each participant. The authors used this model to provide each subject with the maximal amount of EMD stimulus as possible. To take this comment a step further, we also employed 300 total muscle actions performed at 45°/second, which is done in other studies {Lau, 2015 #1615} within similar populations as an approach to successfully induce EMD. One key missing component of our description of the EMD protocol, is the amount of effort by which we required participants. We have clarified this in section 2.9, and also detailed our motivation procedures.
      1. “For all repetitions, participants were instructed to “kick out” as hard as possible against the lever-arm during each of the aforementioned eccentric muscle actions. Strong verbal encouragement was supplied by the research team during the EMD exercise protocol for all participants.”

  • In the 'Results' section, some results refer to "condition" in general, but it's not always clear whether CON vs. sMLCT results are being directly compared, or just discussed within a group. Add more between-group comparisons, especially for time points where differences are hypothesised or relevant to your aims.
    1. As described in section 2.12, our model used fixed effects of condition and time. Condition was described as a categorical variable, including only the CON and sMLCT groups. When we discuss the effect of condition, we are describing said effect within the context of our model. Therefore, as there are only two levels within the “condition” factor, any said effect simply indicates there is a ‘statistical difference’ between CON and sMLCT. Said another way, the authors use “condition” in this context specifically as a fixed effect within the linear mixed-effect model we describe in our methods section. Having said that, if there are specific instances in which we can be clearer we would be happy to address that upon future review iterations.

  • Although the statistical details are rich, readers may miss the broader implications without interpretive summaries. End each subsection with 1-2 sentences summarising the main trend (e.g. "These results suggest that recovery was faster in sMLCT at early time points for RT100 and RT200, despite non-significant interaction effects.)
    1. This is a very valuable comment, and we appreciate the feedback. We have tried to summarize the main trends within our results section at the end of each subsection to help with the interpretation of the results throughout.

  • Tables 1 and 2 are mentioned, but basic group means and SDs for key variables (e.g. peak torque, RT metrics, pain scores) are not consistently reported in the text. Include brief summary statistics for important outcome variables for context (especially pre/post values where changes are discussed).
    1. Thank you for this comment. The mean ± SE values for all key variables are presented in Figures 1 (muscular strength variables), 3 (Ultrasound muscle characteristics), 5 (subjective variables), 7 (anthropometry), and 9 (vertical jump performance). Because of this, we do not feel it is appropriate to additionally display these values in tabular form, as this would be redundant. Having the same data displayed in tables and figures is typically discouraged in research publications.

  • Some parts (e.g. repeated mention that post72 values weren't significant) could be condensed to avoid repetition. Group results for non-significant changes in a concise format (e.g. 'No significant differences were found at post72 for all RT variables except RT200').
    1. Thank you for this comment as we believe it is valuable. To reduce redundancy, we have tried to implement these changes throughout the results section.

  • There's a mix of reporting rank transformed data vs. raw data. Please clarify the rationale for this distinction (perhaps in the methods) or ensure consistent interpretation guidance.
    1. We appreciate the comment made by the reviewer and have since attempted to ensure provide clarity in section 2.12. Specifically, the authors utilized the rank transformed data for all models. However, in efforts to maximize readability for our future readers, raw data was used in all tables and figures. This approach is common and allows the authors to appropriately analyze the data, while maintaining interpretability. Changes in section 2.12 include:
      1. “As described above, VAS outcomes were rank-transformed due to extreme outliers and other violations of model assumptions when using raw data, especially the normality of residuals. However, raw data are displayed to enhance interpretability.”

  • While the study is presented as novel, the "so what" question is underdeveloped. Suggestion: Add a short paragraph at the end of 4.1 or the beginning of 4.6 to explicitly state the contribution of this research to the field of nutritional ergogenics and muscle recovery. My suggestion: "This study provides fundamental insight into the limited efficacy of sMLCT on muscle recovery in a healthy female population, highlighting the need for population-specific supplementation protocols".
    1. We appreciate this astute observation of this reviewer. We have since tried to incorporate this sentence at the end of section 4.1 to highlight the novel insight of this study.
      1. “Indeed, this study provides critical insight into the potential efficacy of sMLCT on muscle recovery in a healthy female population, while also highlighting the need for additional population-specific investigations.”

  • The study focuses on performance metrics, but biochemical markers (e.g. CK, IL-6) were not measured. Suggestion: In 4.6 you mention this as a limitation, but it could also be used to more strongly frame a future research opportunity. My suggestion: "Given the proposed immunomodulatory and metabolic effects of sMLCT, future work incorporating cytokine profiles or oxidative stress markers could clarify whether there is a discrepancy between physiological processes and performance outcomes".
    1. We appreciate the feedback. We have further tried to expand on the limitation of why future studies may want to use biochemical markers to assess muscle damage.
      1. “In a previous study [39], plasma creatine kinase was used as a biomarker to assess muscle damage following an EMD protocol and significant differences were observed at follow-up days 3-5. As sMLCT may influence immunomodulatory and metabolic processes, future works may assess cytokine profiles or markers of oxidative stress to provide additional insights on the potential discrepancy between any noted physiological and performance metrics.”

  • Much of the justification for sMLCT is based on studies in critical care patients or those on parenteral nutrition, which may not be generalisable. Consider condensing this discussion (lines 406-417) and emphasising why these findings may not apply to young, healthy individuals.
    1. We thank the reviewer for this astute comment. We agree that further clarification on the potential generalizability issue is warranted. We have clarified this within section 4.1 to provide appropriate context to the reader when they are understanding the existing body of literature. Specifically:
      1. “However, these data should be interpreted with caution. While they provide mechanistic insights into the potential action of sMLCTs, the findings may not be generalizable outside of critical care patients or those on parenteral nutrition. With that said, despite the mechanistic evidence available, after undergoing EMD in the present study, there were no significant differences in the response between the sMLCT and the CON groups following the acute exercise bout.”
  • The interpretation of null results needs to be more nuanced. Rather than only stating "no significant effect," explore why the supplement might not have worked in this context. My suggestion: "The lack of significant condition × time interactions may be due to the relatively low baseline inflammation in the population studied, or to insufficient bioavailability or dosage of the active compounds."
    1. Your comment is well taken. We have adjusted our manuscript at the end of section 1 to provide a better interpretation of the lack of significant results.
      1. “The lack of a significant interaction effect may be due to a number of factors including the degree of baseline inflammation present in the sample in addition to the dosage and characteristics of the dietary supplement.”

  • There is little translation of the findings into practical advice for sports scientists or clinicians. Please add a concluding statement in 4.6 or a short new section (4.7 Practical implications). My suggestion: "Until further evidence emerges, practitioners may consider that short-term sMLCT supplementation is unlikely to improve recovery in recreational female athletes following eccentric muscle damage."
    1. Thank you for this comment as we believe that it is valuable. We have tried to add a sentence at the end of section 4.6 to conclude this manuscript with the practical applications of this study
      1. “Until further evidence emerges, clinicians and practitioners may utilize caution when considering the use of short-term sMLCT supplementation in efforts to enhance muscle recovery in recreationally active females.”

  • This manuscript presents valuable findings, but improvements in clarity, depth of discussion and methodological detail would further enhance its impact. I recommend acceptance of the manuscript with minor revisions.
    1. Thank you for not only your positive points of our manuscript but providing helpful recommendations along the way. We believe our manuscript has been substantially improved with your help.

Reviewer 2 Report

Comments and Suggestions for Authors

This manuscript addresses an important and growing area of interest in sports nutrition: the potential role of structured medium- and long-chain triglycerides (sMLCT) in aiding muscular recovery after eccentric exercise-induced muscle damage (EMD). The study is well-conceived in terms of its methodological design—being double-blind, randomized, placebo-controlled—and employs a broad and relevant set of outcome measures, including strength, morphology, subjective soreness, and vertical jump performance. It also reflects commendable attention to experimental rigor, including menstrual cycle control and consistent measurement protocols.

That said, the manuscript's findings are predominantly negative, and the conclusions must be interpreted with substantial caution due to several important methodological and interpretative limitations.

To begin with, the primary limitation is the lack of meaningful effects despite a well-controlled design. The data did not support the hypothesis that sMLCT ingestion would improve recovery from EMD. This lack of effect was consistent across nearly all objective and subjective measures, including peak torque, rapid torque, ultrasound-derived muscle characteristics, vertical jump metrics, and soreness ratings. The only signals suggesting minor potential benefit were isolated timepoint differences in echo intensity and jump velocity—which are not sufficient to support efficacy claims. The paper’s conclusion that sMLCT does not significantly influence recovery is valid but could benefit from being framed more strongly: not only was there a lack of effect, but the design was robust enough to likely detect moderate effects had they existed.

One important conceptual flaw is that the extrapolation from clinical literature on sMLCT (e.g., in parenteral nutrition) to healthy, young, recreationally active females recovering from mechanical muscle damage is tenuous. While the authors attempt to justify this based on mechanisms of immune modulation or protein balance, they do not adequately explain why these mechanisms would translate to enhanced functional recovery in this specific population. The literature cited in support of sMLCT mainly addresses catabolic states or malabsorption—not acute muscle injury in healthy muscle tissue.

The authors correctly highlight that the lack of biomarker data—such as creatine kinase or inflammatory cytokines—is a missed opportunity. The addition of such measures would have strengthened the interpretation of the findings, particularly in relation to mechanisms of action. Moreover, the potential dose or duration of supplementation may have been insufficient to elicit meaningful effects; however, this is only briefly acknowledged and not discussed with enough depth.

From a statistical perspective, while the use of linear mixed models is appropriate, the interpretation of p-values close to significance (e.g., p = 0.05 for peak torque) is at times overly optimistic. The authors should avoid implying trends unless they are substantiated by adjusted models or replicated across multiple timepoints or outcomes. Effect sizes or confidence intervals should be more prominently reported and discussed to contextualize the clinical relevance of observed differences.

In terms of writing and clarity, the manuscript is well-organized and generally readable. However, it suffers from a degree of redundancy, especially in the discussion section where previously stated results are reiterated without adding new interpretation. The authors could improve conciseness by integrating tables and figures more effectively with the text. Additionally, while the authors disclose financial support and potential conflicts of interest transparently, the tone throughout the discussion occasionally borders on advocacy rather than impartial reporting, particularly in defending the theoretical benefits of sMLCT despite the lack of empirical support.

Key recommendations for revision include:

Reframing the discussion more critically to emphasize the study's null findings and their implications.

Providing a more nuanced examination of why sMLCT may not work in this population or context.

Including more detailed effect size data and confidence intervals throughout.

Condensing redundant content and focusing on interpreting the most relevant findings.

Adding a clearer call for replication in different populations or with longer supplementation periods.

Despite the lack of significant outcomes, the study adds value by providing high-quality evidence that helps clarify the limitations of sMLCT as a recovery aid. Its publication could prevent redundant studies and redirect research toward more promising interventions—if framed appropriately as a robust negative trial.

Author Response

Reviewer 2

  • This manuscript addresses an important and growing area of interest in sports nutrition: the potential role of structured medium- and long-chain triglycerides (sMLCT) in aiding muscular recovery after eccentric exercise-induced muscle damage (EMD). The study is well-conceived in terms of its methodological design—being double-blind, randomized, placebo-controlled—and employs a broad and relevant set of outcome measures, including strength, morphology, subjective soreness, and vertical jump performance. It also reflects commendable attention to experimental rigor, including menstrual cycle control and consistent measurement protocols.

That said, the manuscript's findings are predominantly negative, and the conclusions must be interpreted with substantial caution due to several important methodological and interpretative limitations.

  1. Thank you for taking the time and effort to review our work.

  • To begin with, the primary limitation is the lack of meaningful effects despite a well-controlled design. The data did not support the hypothesis that sMLCT ingestion would improve recovery from EMD. This lack of effect was consistent across nearly all objective and subjective measures, including peak torque, rapid torque, ultrasound-derived muscle characteristics, vertical jump metrics, and soreness ratings. The only signals suggesting minor potential benefit were isolated timepoint differences in echo intensity and jump velocity—which are not sufficient to support efficacy claims. The paper’s conclusion that sMLCT does not significantly influence recovery is valid but could benefit from being framed more strongly: not only was there a lack of effect, but the design was robust enough to likely detect moderate effects had they existed.
    1. We thank you for your insights and are glad you agree with our findings. We believe that changes to the manuscript from reviewer 1 and yourself, have indeed more strongly framed the lack of meaningful effect presented by sMLCT in the current investigation.

  • One important conceptual flaw is that the extrapolation from clinical literature on sMLCT (e.g., in parenteral nutrition) to healthy, young, recreationally active females recovering from mechanical muscle damage is tenuous. While the authors attempt to justify this based on mechanisms of immune modulation or protein balance, they do not adequately explain why these mechanisms would translate to enhanced functional recovery in this specific population. The literature cited in support of sMLCT mainly addresses catabolic states or malabsorption—not acute muscle injury in healthy muscle tissue.
    1. We appreciate this comment as we believe it is valuable, Reviewer 1 had a similar concern. To clarify, the existing body of literature on sMLCT in humans is not robust. Therefore, when searching for evidence as to it’s potential impacts on the study’s dependent variables, the authors were limited in manuscript selection. In other words, some of the only available literature in humans happens to be in said clinical populations. It would be inappropriate for our work not to discuss those findings. However, we tend to agree with you and reviewer 1 in that we needed to better guide the future reader in how to appreciate those works in the proper context. To appease both reviewers, we have attempted to clearly and bolsters the context of available, mechanistic data for sMLCT. The following is an example of such bolstering in section 1 (introduction), paragraph 2.
      1. “Although there is limited research on sMLCT in non-clinical populations, the physiological mechanism by which medium- and long-chain fatty acids aid muscle tissue recovery is by promoting a positive nitrogen balance [14, 15] and potential anti-inflammatory properties [15], therefore providing strong rational of the use of sMLCT in athletic settings.”

  • The authors correctly highlight that the lack of biomarker data—such as creatine kinase or inflammatory cytokines—is a missed opportunity. The addition of such measures would have strengthened the interpretation of the findings, particularly in relation to mechanisms of action. Moreover, the potential dose or duration of supplementation may have been insufficient to elicit meaningful effects; however, this is only briefly acknowledged and not discussed with enough depth.
    1. This is a very valuable comment, and we appreciate the feedback. Reviewer 1 had a similar insight, and, in response, we have expanded this thought on a few instances in the middle and end of section 4.6 to further strengthen the interpretation of the limitation within this study for our future readers.
      1. “In a previous study [39], plasma creatine kinase was used as a biomarker to assess muscle damage following an EMD protocol and significant differences were observed at follow-up days 3-5. As sMLCT may influence immunomodulatory and metabolic processes, future works may assess cytokine profiles or markers of oxidative stress to provide additional insights on the potential discrepancy between any noted physiological and performance metrics.”
      2. “The lack of a direct effect based on the condition (i.e., sMLCT, CON) suggests that short-term sMLCT supplementation is unlikely to enhance recovery in recreationally active females following an EMD protocol, within the context of the investigated supplementation and exercise protocol.” “Further research is needed to determine the potential role of sMLCT in enhancing muscular performance and recuperation strategies under different conditions.”

  • From a statistical perspective, while the use of linear mixed models is appropriate, the interpretation of p-values close to significance (e.g., p = 0.05 for peak torque) is at times overly optimistic. The authors should avoid implying trends unless they are substantiated by adjusted models or replicated across multiple timepoints or outcomes. Effect sizes or confidence intervals should be more prominently reported and discussed to contextualize the clinical relevance of observed differences.
    1. Thank you for your compliment on our detailed statistical analyses section. However, due to the nature of how linear mixed effects models deal hierarchical grouping via the random effect, traditional measures of effect size (i.e., partial eta squared) are not recommended to be quantified. The reason being is the interpretation of the effect size, as derived from the LME, becomes difficult and inappropriate, especially when compared to traditional linear models (i.e., ‘ANOVA’), which are mathematically different.

  • In terms of writing and clarity, the manuscript is well-organized and generally readable. However, it suffers from a degree of redundancy, especially in the discussion section where previously stated results are reiterated without adding new interpretation. The authors could improve conciseness by integrating tables and figures more effectively with the text. Additionally, while the authors disclose financial support and potential conflicts of interest transparently, the tone throughout the discussion occasionally borders on advocacy rather than impartial reporting, particularly in defending the theoretical benefits of sMLCT despite the lack of empirical support.

    1. Thank you for this comment, through the revision process, we believe the structure of the discussion has been sufficiently altered to reduce any redundancy. However, if such remains, we would be happy to re-consider this upon future revision if the reviewer would be willing to specify where such areas exist.
    2. Additionally, while we appreciate the reviewer’s point on our tone ‘advocating’ or ‘defending’ the use of sMLCT, we would like to clarify that we are not defending the compound. However, we are attempting to frame this from a science perspective. Specifically, by encouraging the reader to look for additional evidence prior to forming a consensus on this topic – one study may not provide the complete picture of the supplement.
      1. “Until further evidence emerges, clinicians and practitioners may utilize caution when considering the use of short-term sMLCT supplementation in efforts to enhance muscle recovery in recreationally active females.”

Key recommendations for revision include:

  • Reframing the discussion more critically to emphasize the study's null findings and their implications.
    1. Thank you – we believe the aforementioned changes have sufficiently and more critically addressed the implication of the study’s findings.

  • Providing a more nuanced examination of why sMLCT may not work in this population or context.
    1. We appreciate this comment and believe that we have effectively addressed the this concern by adding the following within our manuscript.
      1. “The lack of a significant interaction effect may be due to a number of factors including the degree of baseline inflammation present in the sample in addition to the dosage and characteristics of the dietary supplement.”
      2. “The lack of a direct effect based on the condition (i.e., sMLCT, CON) suggests that short-term sMLCT supplementation is unlikely to enhance recovery in recreationally active females following an EMD protocol, within the context of the investigated supplementation and exercise protocol.”
  • “Until further evidence emerges, clinicians and practitioners may utilize caution when considering the use of short-term sMLCT supplementation in efforts to enhance muscle recovery in recreationally active females.”

  • Including more detailed effect size data and confidence intervals throughout.
    1. As we described above, the use of traditional measures of effect size (e.g., partial eta squared) may not be appropriate when using LMEs. We agree that confidence intervals are important for proper interpretation of each model parameter and have reported each within the study figures.

  • Condensing redundant content and focusing on interpreting the most relevant findings.
    1. As mentioned in a prior comment, we believe a reduction in redundancy and more focused interpretation of the findings has been completed, thanks to comments from both reviewers.
      1. “Until further evidence emerges, clinicians and practitioners may utilize caution when considering the use of short-term sMLCT supplementation in efforts to enhance muscle recovery in recreationally active females.”

  • Adding a clearer call for replication in different populations or with longer supplementation periods.
    1. We appreciate the need for replication work and thank the reviewer for making the point. We have addressed this concern by adding the following:
      1. “The lack of a direct effect based on the condition (i.e., sMLCT, CON) suggests that short-term sMLCT supplementation is unlikely to enhance recovery in recreationally active females following an EMD protocol, within the context of the investigated supplementation and exercise protocol. Further research is needed to determine the potential role of sMLCT in enhancing muscular performance and recuperation strategies under different conditions. Until further evidence emerges, clinicians and practitioners may utilize caution when considering the use of short-term sMLCT supplementation in efforts to enhance muscle recovery in recreationally active females.”

  • Despite the lack of significant outcomes, the study adds value by providing high-quality evidence that helps clarify the limitations of sMLCT as a recovery aid. Its publication could prevent redundant studies and redirect research toward more promising interventions—if framed appropriately as a robust negative trial.
    1. Thank you for your helpful comments. The review process has allowed our manuscript to be strengthened!

Round 2

Reviewer 2 Report

Comments and Suggestions for Authors

Congratulations